# Health Care and Productivity Costs of Non-Fatal Traffic Injuries: A Comparison of Road User Types

**DOI:** 10.3390/ijerph17072217

**Published:** 2020-03-26

**Authors:** Marjolein van der Vlegel, Juanita A. Haagsma, Leonie de Munter, Mariska A. C. de Jongh, Suzanne Polinder

**Affiliations:** 1Department of Public Health, Erasmus MC, University Medical Center Rotterdam, 3015 GD Rotterdam, The Netherlands; 2Department Trauma TopCare, ETZ Hospital, 5022 GC Tilburg, The Netherlands

**Keywords:** health care costs, productivity loss, road traffic injuries, injury

## Abstract

This study aimed to provide a detailed overview of the health care and productivity costs of non-fatal road traffic injuries by road user type. In a cohort study in the Netherlands, adult injury patients admitted to a hospital as a result of a traffic accident completed questionnaires 1 week and 1, 3, 6, 12 and 24 months after injury, including the iMTA Medical Consumption and Productivity Cost Questionnaire. In-hospital, post-hospital medical costs and productivity costs were calculated up to two years after traffic injury. In total, 1024 patients were included in this study. The mean health care costs per patient were € 8200. The mean productivity costs were € 5900. Being female, older age, with higher injury severity and having multiple comorbidities were associated with higher health care costs. Higher injury severity and being male were associated with higher productivity costs. Pedestrians aged ≥ 65 years had the highest mean health care costs (€ 18,800) and motorcyclists the highest mean productivity costs (€ 9000). Bicycle injuries occurred most often in our sample (n = 554, 54.1%) and accounted for the highest total health care and productivity costs. Considering the high proportion of total costs incurred by bicycle injuries, this is an important area for the prevention of traffic injuries.

## 1. Introduction

Globally, road traffic accidents are a leading cause of death for people aged between 5 and 29 years and the 8^th^ leading cause of death for people of all ages with 1.35 million deaths each year. Road traffic accidents also cause up to 50 million non-fatal injuries each year [1].

A systematic review on road traffic fatalities by road user group estimated that 63% of all deaths in high-income countries were among motorized four wheelers, compared to 34% in low-income countries. The distribution of road user accidents varies widely across countries by road user type, age and sex [2]. Another study showed that the incidence and severity of injuries vary according to the type of road user [3]. For example, pedestrians and motorcyclists suffered the most severe injuries and had a greater use of health care in the first three months compared to other road users. 

A widely used measure, that incorporates health care consumption to express the societal impact of diseases or injuries, is cost-of-illness. However, research on cost-of-illness of traffic-related injuries is scarce and only few studies have investigated the differences in these costs between road user types. Two studies conducted in the Netherlands and the US assessed health care and productivity costs of road traffic injury by road user type [4,5]. Health care costs consist of emergency department (ED) visits, hospital stays and diagnostics, but also health care consumption outside of the hospital, for example stays or treatment in a rehabilitation center or homecare. Productivity costs consist of costs associated with productivity loss due to injury also add to the total health care costs related to injury. With regards to health care costs, it should be noted that in the Netherlands, there is a system of obligatory health insurance, provided through private health insurance companies. These companies have to provide a standard package of insured treatments. Both short-term hospitalization and long-term medical care are covered by this mandatory insurance for the whole population.

The findings of the two studies on health care and productivity costs of road traffic injury showed high variability of incidence and health care costs by road user type. However, data of the US study were collected in 2005 and data of the Dutch study were collected 2007. In the Netherlands, since 2007, several changes occurred with regards to the demographics of road users, speed and distance travelled, that may affect the incidence and severity of traffic injuries, and subsequently cost of injury. These changes included an increase in use of bicycles with a pedal-assist electric drive system (e-bikes) and an increase in the distances that are traveled by the elderly, both by bicycle and by car [6,7,8]. These marked changes in a relatively short period of time underline the importance of up-to-date cost estimates by road user type which are currently lacking, yet essential to evaluate existing road safety policies and to provide insight into potential areas for saving costs. 

This study aimed to give an overview of the health care and productivity costs of traffic-related injuries and to compare these costs by six road user types including car occupants, motorcyclists, moped riders, (electric) bicycle riders, pedestrians and unspecified road users, based on data collected from 2015 to 2018.

## 2. Materials and Methods 

### 2.1. Study Setting

This study is linked to the Brabant Injury Outcome Surveillance (BIOS) study. The BIOS study was a prospective longitudinal cohort study with a 24-month follow-up of trauma patients that had been admitted to any of the ten hospital in the Dutch Noord-Brabant region [9]. One of these hospitals is a level I hospital, and the other nine hospitals are level II and level III hospitals. Patients that were aged 18 years or older who were admitted to a ward or an Intensive Care unit (ICU) and survived hospital discharge were included in the BIOS study. Patients who had a pathological fracture, insufficient knowledge of the Dutch language or the absence of a permanent address of residence were excluded from the study. Patients were recruited between August 2015 and November 2016 and were invited to participate at 1 week post-trauma. Eligible patients were included in this study if they completed at least one questionnaire and in-hospital data of the ICU or ward were available. The BIOS study has been approved by the Medical Ethics Committee Brabant (NL50258.028.14).

### 2.2. Patient and Injury Characteristics

In our study, only patients that sustained their injury as a result of a traffic accident were included. A traffic accident was defined as an accident that occurred on a street open to public traffic involving pedestrians, cyclists and vehicles. Road user categories were car occupants (includes passenger cars, taxis, light goods vehicles), motorcyclists, moped riders (includes mopeds, scooters), (electric) bicycle riders, pedestrians and unspecified road users. There was no information available to separate bicyclists and electric bicyclists, therefore they were placed in one category. The unspecified road users included horse-riders, wheelchair users, mobility scooter users and persons who sustained a traffic-related injury but whose road user type was unknown. Data on the type of road user as well as data on injury severity and type of injury were obtained from the Brabant Trauma Registry (BTR) and merged to the BIOS-data. The BTR collects prehospital and hospital data of all trauma patients admitted after presentation to the emergency department at any of the ten hospital in the Dutch Noord-Brabant region.

The overall trauma severity was assessed by the Injury Severity Score (ISS), which ranges from 1 (minor injuries) to 75 (untreatable injury). The ISS was computed using the Abbreviated Injury Scale (AIS) (AIS-90, update 2008) [10] score of the three most severely injured body regions. The AIS scores the location, type and severity of each injury. The highest AIS of these three body regions are squared to calculate the ISS.

Each injury patient that met the inclusion criteria of the study received a postal questionnaire 1 week and 1, 3, 6, 12 and 24 months after injury. A proxy informant completed the self-reported questionnaires if the patients were incapable of completing the questionnaires themselves. All participants and proxy informants signed informed consent for participation in the BIOS study. T1 and T2 included questions on socio-demographic characteristics including age, sex and comorbidities. The following comorbidities were included in this study: heart disease, vascular disease, lung disease, consequences of a stroke, neurological disease, kidney disease, diabetes mellitus, osteoporosis, dementia, psychiatric disorder, herniated disk or other severe back problems, arthritis, rheumatism and cancer. There was also the option to record other chronic diseases that did not fall under the disease categories named above.

### 2.3. Health Care Consumption, Productivity Loss and Cost Estimation

In-hospital medical procedures are obtained from the BTR and detailed information on intramural and extramural care costs are collected with the iMTA Medical Consumption Questionnaire (iMCQ), consisting of 31 questions related to non-disease specific health care consumption [11]. The iMCQ consists of questions related to intramural medical care (e.g., stay or treatment at a medical facility) and extramural health care (e.g., homecare). The iMCQ was included in all questionnaires at 1 week and 1, 3, 6, 12 and 24 months after injury. Costs of transportation to the ED, stay and care at a hospital ward, stay at ICU and diagnostic procedures were part of the in-hospital costs. Stay and care at an institution (e.g., nursing home), home care and contact with practitioners (e.g., general practitioner, physiotherapist) were part of the post-hospital costs. Diagnostic procedures were retrieved from the hospital registrations. All unit costs were retrieved from a cost-reference manual, presented in Table A1 [12], except for unit costs of diagnostics, which were retrieved from hospital price lists, previous research and the Dutch Healthcare Authority (NZa) [13,14,15,16,17,18,19,20]. Health care costs were calculated by multiplying health care use per period with cost per unit.

Indirect non-medical costs are defined as costs outside of health care, as a value of productivity loss due to injury. Information on productivity loss at work was collected with the iMTA Productivity Cost Questionnaire (PCQ) [21], consisting of questions on both the absence of work due to injury and on being present at work after injury but being less productive than before the injury. The costs of absenteeism were determined with the friction costs method. The friction costs method assumes that productivity loss is confined to the period needed to find a replacement for the absent employee. Productivity loss longer than the friction period (85 d) was valued equal to 85 d [12]. The average wage rates per gender can be found in the Table A1. If the number of hours a participant worked per week were missing, the national mean based on gender was used, 36 h per week for males and 26 h per week for females [22]. The costs of productivity loss were determined by multiplying the total number of hours of work missed with the hourly wage rate. The productivity costs were determined for the working population (patients aged 18–67). For the calculation of total mean costs (in-hospital costs, post-hospital costs, productivity costs), productivity costs for patients aged > 67 and patients without paid employment were equal to 0. Costs were inflation-adjusted to 2017 euro using consumer price index rates. The average annual income in the Netherlands was € 37,000 in 2017 and the health expenditure per capita in 2017 was € 3791 [23]. 

### 2.4. Statistical Data Analysis

Differences between the responders and non-responders regarding demographic and injury-related characteristics were tested with a Mann-Whitney test (continuous) or chi-square test of homogeneity (categorical). Generalized linear models with gamma distribution and log-link function were used to determine the association between age, gender, injury severity and comorbidities, and health care costs and productivity costs for the total traffic injury population. The model for productivity costs only included the patients that had paid employment and reported productivity loss. The relative difference in mean costs (exp [parameter estimate]) with a 95% Confidence Interval (CI) was reported. Age was categorized as 18–34 years, 35–64 years and ≥ 65 years and the ISS was categorized as 1–3, 4–8, 9–15 and ≥16. The number of comorbidities was categorized as having no comorbidities, one comorbidity or two or more comorbidities. A p-value <0.05 was considered statistically significant. All analyses were conducted with SPSS version 24.0 (statistical package for social sciences, Chicago, IL, USA).

## 3. Results

### 3.1. Characteristics by Road User Type

In total, 1024 out of 2281 people admitted to a hospital because of a traffic-related injury (44.9%) were included in the BIOS study. The distribution of road user groups differed significantly between responders and non-responders (chi^2^ = 73.1, p < 0.001) (Table A2). The majority of respondents were injured during cycling (n = 554; 54.1%) against 41.9% (n = 527) of non-responders. The age of responders was statistically significantly (p < 0.001) higher in responders (56.9 years) than in non-responders (51.0 years). The proportion of female and male responders was not statistically significantly different than non-responders (chi^2^ = 3.3, p = 0.07). The characteristics of participants by road user type are shown in Table 1. Most of the traffic-related injuries occurred in men (n = 598, 58.4%) and nearly all injured motorcyclists were male (n = 37, 97.4%). Pedestrians were the only road users that were predominantly female (n = 22, 61.1%). The median age of the total traffic population was 59 years and median age by road user ranged from 49 years (car occupants) to 63 years (bicyclists). Pedestrians had the highest injury severity, with a median ISS of 7.5 (IQR = 4.0–10.0) and had the longest hospital stay (median 6.5 d). Car occupants had the lowest median ISS of 4 and the shortest hospital stay (median 3.0 d). 

### 3.2. Health Care and Productivity Costs by Road User Type

Table 2 provides a detailed overview of in-hospital, post-hospital and productivity costs by road user type. Total health care costs up to two years after injury ranged from € 10,000 for (electric) bicyclists to € 18,000 for pedestrians. In-hospital costs were higher than post-hospital costs for all road users, except pedestrians. Post-hospital costs for pedestrians, with mean costs of € 9600 per patient, were more than twice as high as for other road users with mean costs ranging from € 3200 for (electric) bicyclists to € 4500 for motorcyclists. 

Mean productivity costs were highest for motorcyclists (€ 9000) and lowest for (electric) bicyclists (€ 5200) and unspecified road users (€ 5100). The number of injuries per road user type differed widely. For example, injuries of pedestrians occurred in 3.5% (n = 36) of the cases in this study, while bicycle injuries were most common (n = 554 (54.1%)). 

Although mean health care and productivity costs were lower for (electric) bicyclists compared to other road users, their total health care and productivity costs were by far the highest because of the high number of injuries due to traffic accidents involving bicyclists, as shown in Figure 1a,b. Reversely, mean health care costs were highest for pedestrians and mean productivity costs were highest for motorcyclists, whereas the total health care and productivity costs were lowest compared to other road users. 

### 3.3. Predictors of Health Care and Productivity Costs

Health care costs were 1.21 (1.06–1.34) times higher for females than males (Table 3). Yet, productivity costs were significantly lower for females than males (0.68 [0.58–0.80]). Higher injury severity was found to be associated with higher health care and higher productivity costs compared to low injury severity. Higher age and ≥2 comorbidities were also associated with higher health care costs compared to respectively lower age and no comorbidities. Age and having comorbidities were not found to be associated with higher productivity costs.

### 3.4. Health Care Costs by Road User Type and Age Groups

Pattern of costs by age group vary widely for the different road type users. Within the road user categories, health care cost differed across age groups (Figure 2). Post-hospital costs were highest in pedestrians age 35 and older and in motorcyclists aged 18–34. The mean health care costs for motorcyclists in the 18–34 age group were remarkably high, partly attributable to one case with severe injury (ISS 26) with high in-hospital and post-hospital health care consumption. Productivity costs were especially high for motorcyclists in the 35–64 age group and for younger pedestrians in the age group of 18–34 years. Total costs of (electric) bicyclists did not vary much across age groups. However, post-hospital costs were highest in the ≥65 years group. Overall, patients ≥65 years had the highest mean health care costs and road users in the 35–64 age group had the highest mean productivity costs.

## 4. Discussion

This study estimated the costs of traffic injury of six road user types. On an individual level, health care costs were highest for older pedestrians and productivity costs were highest for younger motorcyclists. Bicycle injuries occurred most frequently in our sample and accounted for the highest total health care and productivity costs in our study sample. 

Our study showed that female sex, increasing age and injury severity and having multiple comorbidities is associated with higher health care costs for traffic-related injuries. This corresponds to the results of a previous study from the Netherlands from 2017, that also showed higher health care costs for females and older adults for the entire injury population, but did not specifically show which factors were associated with traffic-related injury [5]. The study from the US only reported the total of medical and productivity costs and used a different approach (human capital) to calculate productivity losses [4]. They showed that in total, females and older adults had lower costs compared to males and young adults. This was primarily driven by the productivity costs which are higher for males because of higher average wages. The distribution of road user types was also different to our study. For example, the proportion of bicyclists in our study is very high compared to the study from the US, as would be expected since cycling levels in the Netherlands are more than ten times higher than in the US [24].

The distribution of road user types in our study was similar to the findings of the previous study from the Netherlands [5]. They also found that more than half of the patients were bicyclists and showed that motorcyclists and pedestrians had the highest mean costs. The mean health care costs of the total traffic population were twice as low as the health care costs of the traffic population in our study. This can be explained by differences in injury severity of the study population. The earlier study from the Netherlands used data from the Dutch Injury Surveillance System (LIS) and included patients that were treated at an ED and subsequently discharged to the home environment as well as patients that were admitted to a hospital. 

A strength of this study is that it provided a detailed overview of in-hospital, post-hospital and productivity costs for the total traffic injury population and by road user type. Therefore, we were able to compare these components across road user types. There are several limitations to this study. Firstly, people with minor injuries treated by their general practitioner or at the ED followed by discharge to the home environment were not included in this study. Minor traffic injuries were therefore not represented in this study and therefore it is probable that the mean costs per patient of the entire traffic injury population in the Netherlands are lower than presented in this study. Secondly, the patients included in this study differed by age and road user type from the patients who did not participate. It should be taken into account that the proportion of older patients and patients with bicycle injuries was higher in respondents compared to non-respondents. Since older age was found to be associated with higher health care costs, mean health care costs of the entire traffic injury population in Brabant, the Netherlands, may be lower than reported in this study. Thirdly, in the calculation of productivity costs, the average wage rates of males and females in the Netherlands were used. Since we did not use age-specific wage rates, it is possible that productivity costs were underestimated for prime working age adults and overestimated for young adults and adults approaching retirement age [25]. 

Productivity costs are based on Dutch capital incomes and cannot be directly compared to the costs in other countries. However, the magnitude of cost of traffic injuries and the variation between road user is useful information for other countries.

More than 50% of all traffic injuries in our sample were sustained by (electric) bicyclists. Therefore, these road users carry a high proportion of the health care costs related to traffic injuries. In the Netherlands, there is a cycling culture among both young and old and across social economic status groups [26]. A study on bicycle-related traumatic brain injury showed that between 1998 and 2012, bicycle use increased by 14% in the Netherlands. However, this increase was more than 50% for adults aged 55 years and older [27]. Additionally, there is a rapid increase in the use of electric bicycles in the Netherlands, especially by older adults [8,28]. Nearly one-third of all fatal bicycle accidents are e-bike accidents with a high mortality and morbidity rate for older adults [29]. Our study shows a high rate of non-fatal bicycle injuries in older adults, since nearly half of all bicycle injuries were sustained by patients ≥65 years. In this study we were not able to investigate injuries separately for bicyclists and electric bicyclists. With the popularity of e-bike use, the increase in e-bike use by older adults and the risky riding behavior of e-bike riders [30] it is important to focus on this specific group in future research. 

## 5. Conclusions

This study estimated both the health care and productivity costs of traffic injuries and compared these costs between road user types. Being female, older age, with higher injury severity and having multiple comorbidities were associated with higher health care costs. Higher injury severity and being male were associated with higher productivity costs. Both health care costs and productivity costs varied wildly across road user type and age groups. Older pedestrians had the highest health care costs and motorcyclists had the highest productivity costs. However, more than half of all injuries were due to a bicycle accident, resulting in the highest total health care and productivity costs. Therefore, (electric) bicyclists are a priority in the prevention of traffic injuries in the Netherlands. 

## Figures and Tables

**Figure 1 ijerph-17-02217-f001:**
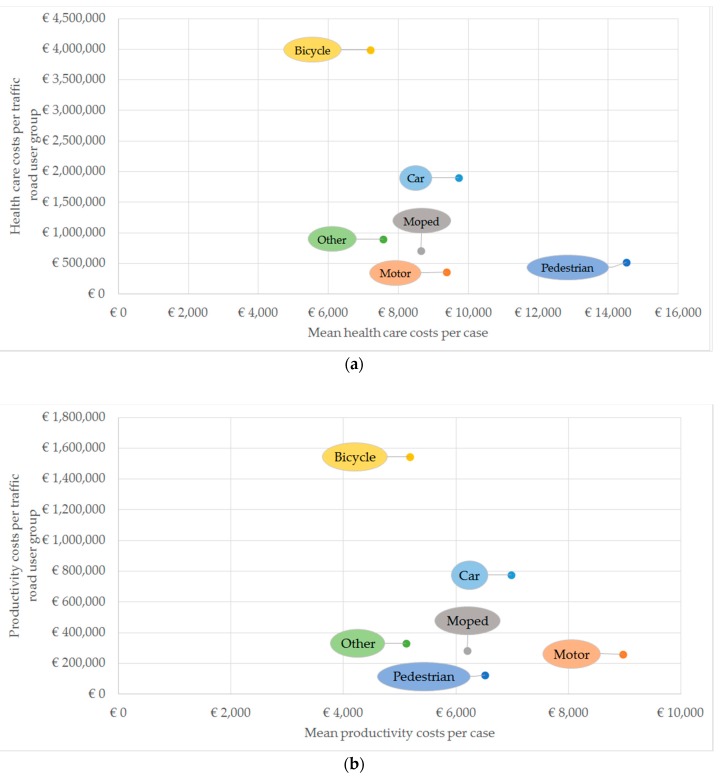
(**a**) Mean health care costs per case and total productivity costs for 6 road type users, 2017 EUR; (**b**) Mean productivity costs per case and total productivity costs for 6 road type users, 2017 EUR.

**Figure 2 ijerph-17-02217-f002:**
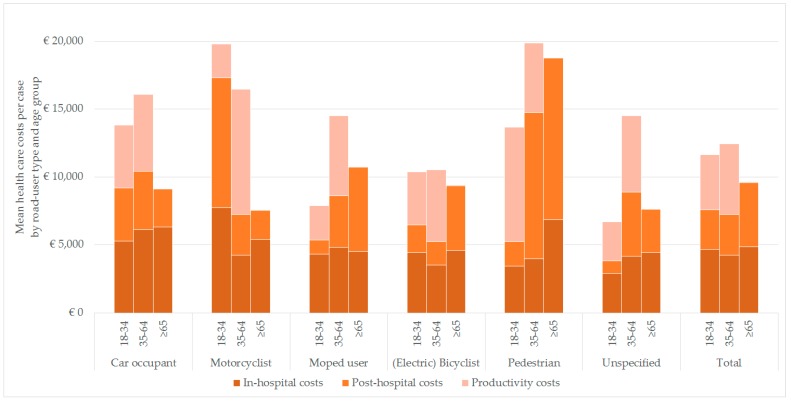
Mean health care costs by road user type and age group, 2017 EUR.

**Table 1 ijerph-17-02217-t001:** Characteristics by road traffic user.

Characteristic	Car Occupant	Motorcyclist	Moped rider	(Electric) Bicyclist	Pedestrian	Unspecified ^4^	Total
N	195 (19)	38 (3.7)	82 (8)	554 (54.1)	36 (3.5)	119 (11.6)	1024
Sex, n (%)							
Male	120 (61.5)	37 (97.4)	53 (64.6)	307 (55.4)	14 (38.9)	67 (56.3)	598 (58.4)
Female	75 (38.5)	1 (2.6)	29 (35.4)	247 (44.6)	22 (61.1)	52 (43.7)	426 (41.6)
Age							
Median (IQR)	49.0 (31.0–64.0)	51.0 (35.0–59.3)	55.0 (40.8–67.3)	63.0 (51.0–71.3)	59.0 (35.8–77.0)	60.0 (47.0–70.0)	59.0 (45.0–70.0)
18–34 years, n (%)	57 (29.2)	8 (21.1)	15 (18.3)	42 (7.6)	8 (22.2)	19 (16)	149 (14.6)
35–64 years, n (%)	90 (46.2)	26 (68.4)	42 (51.2)	257 (46.4)	11 (30.6)	53 (44.5)	479 (46.8)
>65 years, n (%)	48 (24.6)	4 (10.5)	25 (30.5)	255 (46.0)	17 (47.2)	47 (39.5)	396 (38.7)
Type of injury ^1^							
Pelvic injury, n (%)	19 (9.7)	5 (13.2)	9 (11)	46 (8.3)	6 (16.7)	10 (8.4)	95 (9.3)
Hip fracture, n (%)	4 (2.1)	0 (0)	7 (8.5)	91 (16.4)	7 (19.4)	8 (6.7)	117 (11.4)
Tibia, complex foot or femur fracture, n (%)	25 (12.8)	12 (31.6)	15 (18.3)	62 (11.2)	11 (30.6)	18 (15.1)	143 (14.0)
Shoulder and upper arm injury, n (%)	21 (10.8)	7 (18.4)	16 (19.5)	81 (14.6)	5 (13.9)	10 (8.4)	140 (13.7)
Radius, ulna or hand fracture, n (%)	14 (7.2)	8 (21.1)	8 (9.8)	40 (7.2)	3 (8.3)	7 (5.9)	80 (7.8)
Head injury, n (%)	83 (42.6)	6 (15.8)	36 (43.9)	242 (43.7)	16 (44.4)	44 (37.0)	427 (41.7)
Facial injury, n (%)	8 (4.1)	2 (5.3)	11 (13.4)	66 (11.9)	3 (8.3)	9 (7.6)	99 (9.7)
Thoracic injury, n (%)	21 (10.8)	6 (15.8)	7 (8.5)	28 (5.1)	3 (8.3)	4 (3.4)	69 (6.7)
Rib fracture, n (%)	65 (33.3)	10 (26.3)	20 (24.4)	69 (12.5)	3 (8.3)	21 (17.6)	188 (18.4)
Abdominal injury, n (%)	22 (11.3)	4 (10.5)	7 (8.5)	10 (1.8)	2 (5.6)	1 (0.8)	46 (4.5)
Stable vertebral fracture or disc injury, n (%)	22 (11.3)	5 (13.2)	8 (9.8)	32 (5.8)	3 (8.3)	10 (8.4)	80 (7.8)
Injury Severity ^2^							
Median (IQR)	4.0 (2.0–9.0)	5.5 (4.0–10.0)	6.0 (4.0–9.3)	5.0 (4.0–9.0)	7.5 (4.0–10.0)	4.0 (4.0–9.0)	5.0 (3.0–9.0)
ISS 1–3, n (%)	88 (45.1)	6 (15.8)	18 (22.0)	133 (24.0)	7 (19.4)	26 (21.8)	278 (27.1)
ISS 4–8, n (%)	53 (27.2)	17 (44.7)	32 (39.0)	202 (36.5)	11 (30.6)	56 (47.1)	371 (36.2)
ISS 9–15, n (%)	29 (23.7)	9 (23.7)	21 (23.6)	186 (33.6)	14 (38.9)	29 (24.4)	288 (28.1)
ISS ≥16, n (%)	25 (12.8)	6 (15.8)	11 (13.4)	33 (6.0)	4 (11.1)	5 (4.2)	84 (8.2)
Days admitted to hospital							
Median (IQR)	3.0 (2.0–6.0)	4.0 (2.0–11.0)	4.0 (2.0–8.3)	4.0 (2.0–6.0)	6.5 (2.0–11.5)	3.0 (2.0–6.0)	3.0 (2.0–7.0)
Employment, n (%)^3^							
Paid employment	104 (53.3)	30 (78.9)	36 (43.9)	246 (44.4)	14 (38.9)	49 (41.2)	479 (46.8)
No paid employment	28 (14.4)	3 (7.9)	19 (23.2)	87 (15.7)	6 (16.7)	26 (21.8)	169 (16.5)
Retired	45 (23.1)	3 (7.9)	20 (24.4)	199 (35.9)	15 (41.7)	37 (31.1)	319 (31.2)
Comorbidity, n (%)							
No comorbidities	111 (56.9)	29 (76.3)	35 (42.7)	250 (45.1)	21 (58.3)	54 (45.4)	500 (48.8)
1 comorbidity	58 (29.7)	7 (18.4)	23 (28.0)	163 (29.4)	10 (27.8)	39 (32.8)	300 (29.3)
2 or more comorbidities	26 (13.3)	2 (5.3)	24 (29.3)	141 (25.5)	5 (13.9)	26 (21.8)	224 (21.9)

^1^ Percentages add up to more than 100% as respondents can have multiple injuries; ^2^ ISS score of 3 cases (0.3%) in the Unspecified group are missing; ^3^ Unknown employment for 57 (5.6%) cases; ^4^ Includes for example accidents with horse carriages, mobility scooters, wheelchairs and unspecified accidents.

**Table 2 ijerph-17-02217-t002:** Mean (SD) health care and productivity costs of traffic-related injuries by type of road accident, 2017 EUR.

Type of Costs	Car Occupant	Motorcyclist	Moped Rider	(Electric) Bicyclist	Pedestrian	Unspecified ^3^	Total
N (%)	195 (19)	38 (3.7)	82 (8)	554 (54.1)	36 (3.5)	119 (11.6)	1024
In-hospital costs	5900 (8900)	5100 (5100)	4600 (3900)	4100 (3500)	5200 (3300)	4100 (2700)	4500 (5100)
Post-hospital costs	3900 (9100)	4500 (8800)	4100 (9400)	3200 (9700)	9600 (14,500)	3700 (11,900)	3700 (10,100)
Productivity costs ^1^	7000 (7300)	9000 (7700)	6200 (7700)	5200 (6000)	6500 (6800)	5100 (6700)	5900 (6700)
Total costs ^2^	13,700 (17,700)	16,200 (15,000)	12,100 (13,700)	10,000 (12,100)	18,000 (16,800)	10,400 (14,500)	11,400 (14,100)

Costs are rounded to € 100; ^1^ Productivity costs are based on data on population aged 18–67 years with paid employment (n = 568); ^2^ Productivity costs are set to 0 for patients aged >67 years and older and patients without paid employment for the calculation of total mean costs; ^3^ Includes for example accidents with horse carriages, mobility scooters, wheelchairs and unspecified accidents.

**Table 3 ijerph-17-02217-t003:** Unadjusted and adjusted generalized linear models for health care costs and productivity costs.

Characteristic	Health Care Costs			Productivity Costs ^3^		
	Number of Cases N = 1021	Unadjusted Exp(B)	Adjusted ^4^ Exp(B)	Number of cases; N = 394	Unadjusted Exp(B)	Adjusted ^4^ Exp(B)
Sex						
Male (ref)	597			247		
Female	424	1.20 (1.06–1.34)	1.21 (1.09–1.34)	147	0.67 (0.57–0.79)	0.68 (0.58–0.80)
Age						
18–34 years (ref)	148			75		
35–64 years ^1^	478	0.96 (0.80–1.14)	1.21 (1.04–1.42)	313	1.04 (0.84–1.30)	1.01 (0.81–1.24)
>65 years	395	1.27 (1.06–1.51)	1.63 (1.37–1.93)	6	-	-
Injury Severity ^2^						
ISS 1–3 (ref)	278			121		
ISS 4–8	371	1.71 (1.50–1.95)	1.64 (1.45–1.87)	143	1.64 (1.35–2.01)	1.55 (1.27–1.89)
ISS 9–15	288	2.51 (2.19–2.89)	2.34 (2.04–2.67)	101	1.89 (1.51–2.35)	1.79 (1.44–2.22)
ISS ≥16	84	5.28 (4.30–6.48)	6.12 (5.00–7.50)	29	2.44 (1.74–3.40)	2.32 (1.67–3.23)
Comorbidity						
No comorbidities (ref)	498			264		
1 comorbidity	300	1.29 (1.12–1.47)	1.11 (0.98–1.25)	89	1.06 (0.86–1.30)	1.04 (0.86–1.27)
2 or more comorbidities	223	1.32 (1.14–1.54)	1.19 (1.04–1.37)	41	0.93 (0.70–1.24)	1.02 (0.78–1.34)

^1^ For the productivity costs model, this category (34–64 years) also includes the ≥ 65 years category (6 cases with productivity costs); ^2^ ISS scores of 3 cases (0.3%) are missing. These cases are not included in the analysis; ^3^ only included the patients that had paid employment and reported productivity loss; ^4^ adjusted for all other variables presented in the table.

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
