# Peer review of "Health Care and Productivity Costs of Non-Fatal Traffic Injuries: A Comparison of Road User Types"

_ijerph, 2020, doi:10.3390/ijerph17072217_

Round 1

Reviewer 1 Report

Caros autores,

Gostaria de parabenizar os pesquisadores pelo estudo, cujo tema é relevante para que as políticas de segurança possam ser implementadas. Descrevo abaixo algumas contribuições para melhorar a publicação.
1- A introdução é breve e pode contextualizar mais com dados referentes à região estudada (Holanda).
2- Critérios adequados de inclusão, exclusão e amostra.
3- O estudo foi realizado com dados de 1 dos 10 hospitais da região. Descreva se este foi um hospital de referência para os feridos.
4- Breve discussão, precisávamos explorar mais os resultados. Os autores poderiam fazer considerações sobre as melhorias nas políticas de segurança pública, conforme proposto na justificativa.
5- Parabenizo as limitações importantes, relevantes e que devem ser consideradas na interpretação das análises, mas que foram respeitadas com respeito pelos autores.

Author Response

Response to Reviewer 1 Comments

We would like to thank the reviewer for reading our article and providing helpful feedback.

Point 1: The introduction is brief and can contextualize more with data referring to the studied region (Netherlands).

Response 1: We would like to thank the reviewer for pointing this out. To provide more information on the Dutch health care system a description is added in the introduction at line 47:

“With regards to health care costs it should be noted that in the Netherlands, there is a system of obligatory health insurance, provided through private health insurance companies. These companies have to provide a standard package of insured treatments. Both short-term hospitalization and long-term medical care are covered by this mandatory insurance for the whole population..”

Point 2: The study was carried out with data from 1 of the 10 hospitals in the region. Describe whether this was a referral hospital for the injured.

Response 2: In order to clarify, this study was carried out with data from all 10 hospital in the region of Brabant, the Netherlands, not only one of the hospitals. This is described in the methods starting at line 71. To clarify ‘any’ is used instead of ‘one’.

“The BIOS study was a prospective longitudinal cohort study with a 24-month follow-up of trauma patients that had been admitted to any of the ten hospital in the Dutch Noord-Brabant region [9].”

To further specify, the following sentence was added at line 71.

“One of these hospitals is a level I hospital, the other nine hospitals are level II and level III hospitals.”

Also at line 92 ‘any’ is used instead of ‘one’.

Point 3: Brief discussion, we needed to explore the results further. The authors could make considerations about improvements in public security policies, as proposed in the justification.

Response 3: In the introduction we identified an increase of bicycle and car use by elderly (as talked about in the introduction line 55-62). We considered this together with our results in the discussion line 256-266 and therefore recommended in the conclusion that (electric) bicyclists are a priority are for prevention.

Reviewer 2 Report

Manuscript is well written, easy to follow and comprehensive.

Table A2 bottom, below Type of injury needs only formatting for 'or femur fracture' remove or? and place Femur with capital f?, etc. for remaining categories

Author Response

Response to Reviewer 2 Comments

We would like to thank the reviewer for reading our article and providing helpful feedback.

Point 1: (x) English language and style are fine/minor spell check required

Response 1: We thank the reviewer for pointing this out. We checked the whole manuscript on grammar. The following minor changes were made:

Line 15: ‘Completed’ instead of ‘filled out’

Line 37: removed ‘they found that’

Line 38: ‘compared to’ instead of ‘than’

Line 43: removed ‘the’ before health care and productivity costs

Line 52: ‘The findings of the two studies on health care and productivity costs of road traffic injury  showed high variability of incidence and health care costs by road user type’ instead of The findings of these two studies showed high variability of incidence and health care costs by road user type.

Line 73: ‘who’ instead of ‘that’

Line 75: ‘or’ instead of ‘and’

Line 89: ‘who’ instead of ‘that’

Line 170: ‘Table 2 provides’ instead of ‘Table 2 gives’

Line 202: ‘Pattern of costs by age group vary widely for the different road user types’ instead of ‘There is a high variance in pattern of costs by age group for the different road type users.’

Line 189 and line 212: The thousand separator in the figures 1 a and b and figure 2 is changed from . point to , comma.

Point 2: Table A2 bottom, below Type of injury needs only formatting for 'or femur fracture' remove or? and place Femur with capital f?, etc. for remaining categories.

Response 2: We thank the reviewer for this suggestion. Table A2 (line 291) is formatted so each type of injury is on one line, which makes it more clear that femur fracture is part of the group of injuries: Tibia, complex foot or femur fracture. 

Reviewer 3 Report

Thank you for inviting me to review the article entitled "Health care and productivity costs of non-fatal traffic injuries: a comparison of road user types" (ijerph-739733). The article is generally well-written.

Since health care and productivity costs are largely dependent on the local healthcare system and personal incomes, the authors may adjust their findings accordingly. For example, as the data in Table 2, it would be more informative should the authors provide an index that was adjusted to the capital incomes of the surveyed area. The cost index, rather than net costs, would be a better way to compare among different areas or countries. Otherwise, the readers may not have an idea of the number of costs.

In the section of Methods, the authors may define "comorbidities" that were included in the analysis.

The authors divided the patients into several groups by age to compare their overall costs. Is the productivity cost per day of the different age groups the same?  Or should it be adjusted by age? Since the age distribution is varied in different types of traffic accidents, the authors may reconsider the calculation of the total cost based on the concept of age group.

Author Response

Response to Reviewer 3 Comments

We would like to thank the reviewer for reading our article and providing helpful feedback.

Point 1: Since health care and productivity costs are largely dependent on the local healthcare system and personal incomes, the authors may adjust their findings accordingly. For example, as the data in Table 2, it would be more informative should the authors provide an index that was adjusted to the capital incomes of the surveyed area. The cost index, rather than net costs, would be a better way to compare among different areas or countries. Otherwise, the readers may not have an idea of the number of costs.

Response 1: In this paper we wanted to provide an overview of the health care and productivity costs of traffic related injury in Brabant, the Netherlands. All costs are adjusted for the GDP to 2017 EURO. In the article we added the following sentence in the methods to clarify:

Line 136: “Costs were inflation-adjusted to 2017 euro using consumer price index rates.”

In this way, costs can be compared among different areas or countries if those costs are converted to 2017 EURO. 

We added the following paragraph to the discussion to give some information on comparing these costs with the costs in other countries.

Line 253: Productivity costs are based on Dutch capital incomes and cannot be directly compared to the costs in other countries. However, the magnitude of cost of traffic injuries and the variation between road user is useful information for other countries.

Point 2: In the section of Methods, the authors may define "comorbidities" that were included in the analysis.

Response 2: We thank the reviewer for this valuable suggestion. 14 specific types of comorbidities were included in this study, additionally participants had the option to write down other chronic diseases if their disease was not named. A sentence including these conditions is added at line 104.

“The following comorbidities were included in this study: heart disease, vascular disease, lung disease, consequences of a stroke, neurological disease, kidney disease, diabetes mellitus, osteoporosis, dementia, psychiatric disorder, herniated disk or other severe back problems, arthritis, rheumatism and cancer. There was also the option to record other chronic diseases that did not fall under the disease categories named above”

Point 3: The authors divided the patients into several groups by age to compare their overall costs. Is the productivity cost per day of the different age groups the same? Or should it be adjusted by age? Since the age distribution is varied in different types of traffic accidents, the authors may reconsider the calculation of the total cost based on the concept of age group.

Response 3: We thank the reviewer for this valuable suggestion. Productivity costs were calculated based on the ‘Dutch guideline to perform economic evaluation in health care’[1], which provides mean labour costs for men and women. This is a sufficient and widely used method to present mean labour costs. There is no data available to adjust these costs by age using this Dutch guideline.

We do acknowledge however, that this is a limitation. Therefore we added the following sentence in the discussion:

Line 250: Thirdly, in the calculation of productivity costs the Dutch average wage rates of males and females was used. We did not determine the wage rates by age group.

  1. Hakkaart-van Roijen, L., et al., Kostenhandleiding. Methodologie van kostenonderzoek en referentieprijzen voor economische evaluaties in de gezondheidszorg. In opdracht van Zorginstituut Nederland. Geactualiseerde versie, 2015.

Round 2

Reviewer 3 Report

Thank you for inviting me to review the revised version of the manuscript "Health care and productivity costs of non-fatal traffic injuries: a comparison of road user types" (ijerph-739733).

The authors had already responded to the questions and suggestions from the prior reviewers. I would like to suggest the following:
1. In Line 136: “Costs were inflation-adjusted to 2017 euro using consumer price index rates.” The authors can add the average annual income and health expense in the study area so that the readers can have an idea of the relative costs when they apply the study results in other areas.
2. In Line 250: "Thirdly, in the calculation of productivity costs the Dutch average wage rates of males and females was used. We did not determine the wage rates by age group." The authors should elaborate on the limitations by giving more detailed information that how NOT determining the wage rates by age group may confound the interpretation of the study results.

The manuscript may require English language and style checks since minor errors are noted in several paragraphs.  

Author Response

Response to Reviewer Comments

We would like to thank the reviewer for reading our article and providing helpful feedback.

Point 1: In Line 136: “Costs were inflation-adjusted to 2017 euro using consumer price index rates.” The authors can add the average annual income and health expense in the study area so that the readers can have an idea of the relative costs when they apply the study results in other areas.

Response 1: We would like to thank the reviewer for this suggestion. To provide the extra information the following sentence is added:

Line 137: “Costs were inflation-adjusted to 2017 euro using consumer price index rates. The average annual income in the Netherlands was €37.000 in 2017 and the health expenditure per capita in 2017 was €3.791.”

Point 2: In Line 250: "Thirdly, in the calculation of productivity costs the Dutch average wage rates of males and females was used. We did not determine the wage rates by age group." The authors should elaborate on the limitations by giving more detailed information that how NOT determining the wage rates by age group may confound the interpretation of the study results.

Response 2: We thank the reviewer for pointing this out. We added the following information:

Line 250: “Thirdly, in the calculation of productivity costs the average wage rates of males and females in the Netherlands was used. Since we did not use age-specific wage rates, it is possible that productivity costs were underestimated for prime working age adults and overestimated for young adults and adults approaching retirement age [1].  

Point 3: The manuscript may require English language and style checks since minor errors are noted in several paragraphs. 

Response 3: Thank you for this suggestion. We did a spelling check after the first review round. We are open for other text suggestions. Additionally we changed the following:

Line 161: ‘chi2’ instead of ‘chi2’

Line 187: addition of ‘costs’ after in-hospital and post-hospital.

Line 206: ‘aged’ instead of ‘age’

Line 206: ‘The mean health care costs for motorcyclists in the 18-34 age group’ instead of ‘The mean health care costs for 18-34 years motorcyclists’

Line 209: ‘35-64 age group’ instead of ‘ age group 35-64 years’

Line 212: ‘age’ instead of ‘year’

Line 265: ‘one-third’ instead of ‘one third’

  1. Connolly, M.P., et al., A comparison of average wages with age-specific wages for assessing indirect productivity losses: analytic simplicity versus analytic precision. The European journal of health economics : HEPAC : health economics in prevention and care, 2017. 18(6): p. 697-701.